# Reduction in the Migration Activity of Microglia Treated with Silica-Coated Magnetic Nanoparticles and their Recovery Using Citrate

**DOI:** 10.3390/cells11152393

**Published:** 2022-08-03

**Authors:** Tae Hwan Shin, Da Yeon Lee, Yong Eun Jang, Do Hyeon Kwon, Ji Su Hwang, Seok Gi Kim, Chan Seo, Man Jeong Paik, Ju Yeon Lee, Jin Young Kim, Seokho Park, Sung-E Choi, Shaherin Basith, Myeong Ok Kim, Gwang Lee

**Affiliations:** 1Department of Physiology, Ajou University School of Medicine, 206 World Cup-ro, Suwon 16499, Korea; catholicon@ajou.ac.kr (T.H.S.); ekdus93@ajou.ac.kr (D.Y.L.); jys613k@naver.com (Y.E.J.); dohyeon248@ajou.ac.kr (D.H.K.); se777@hanmail.net (S.-E.C.); shaherinb@gmail.com (S.B.); 2Department of Molecular Science and Technology, Ajou University, 206 World Cup-ro, Suwon 16499, Korea; 456547@naver.com (J.S.H.); rlatjrrl9977@naver.com (S.G.K.); 3College of Pharmacy, Sunchon National University, 255 Jungang-ro, Suncheon 57922, Korea; oppschen@naver.com (C.S.); paik815@sunchon.ac.kr (M.J.P.); 4Research Center of Bioconvergence Analysis, Korea Basic Science Institute, 162 Yeongudanji-ro, Cheongju 28119, Korea; jylee@kbsi.re.kr (J.Y.L.); jinyoung@kbsi.re.kr (J.Y.K.); 5Department of Biomedical Science, Graduate School of Ajou University, 206 World Cup-ro, Suwon 16499, Korea; gdj3315@ajou.ac.kr; 6Division of Life Science and Applied Life Science (BK21 FOUR), College of Natural Sciences, Gyeongsang National University, 501 Jinjudae-ro, Jinju 52828, Korea; mokim@gnu.ac.kr

**Keywords:** nanotoxicology, transcriptomics, proteomics, metabolomics, integrated omics, microglia, migration, exocytosis

## Abstract

Nanoparticles have garnered significant interest in neurological research in recent years owing to their efficient penetration of the blood–brain barrier (BBB). However, significant concerns are associated with their harmful effects, including those related to the immune response mediated by microglia, the resident immune cells in the brain, which are exposed to nanoparticles. We analysed the cytotoxic effects of silica-coated magnetic nanoparticles containing rhodamine B isothiocyanate dye [MNPs@SiO_2_(RITC)] in a BV2 microglial cell line using systems toxicological analysis. We performed the invasion assay and the exocytosis assay and transcriptomics, proteomics, metabolomics, and integrated triple-omics analysis, generating a single network using a machine learning algorithm. The results highlight alteration in the mechanisms of the nanotoxic effects of nanoparticles using integrated omics analysis.

## 1. Introduction

Nanoparticles (NPs) with sizes ranging from 1 to 100 nm are increasingly utilised in the field of neuroscience for the diagnosis and effective treatment of various brain diseases, including brain cancer and neurodegenerative diseases [1,2,3]. Since the advantages of NPs for brain research include efficient delivery to the central nervous system (CNS) via the development of a suitable formulation, minimal side effects, and specific deposition in a targeted region [4,5], a variety of NPs has been developed, such as gold NPs, magnetic nanoparticles, liposomes, dendrimers, carbon nanotubes, polymeric nanoparticles, quantum dots, and silica-coated magnetic nanoparticles containing rhodamine B isothiocyanate dye [MNPs@SiO_2_(RITC)] for the diagnosis, monitoring, and treatment of diseases [6,7,8,9,10]. MNPs@SiO_2_(RITC) consist of a ~9 nm cobalt ferrite (CoFe_2_O_3_) core and a RITC-encompassed silica shell [7]. However, several nanotoxicity-related effects have been observed in terms of immune responses induced by microglia in the brain.

NPs can penetrate the brain by passing the blood–brain barrier (BBB) through receptors and transporters expressed on the endothelial cells of brain capillaries, and the NPs have been used as one of the tools for brain-targeting diagnostics and drug delivery [11,12,13]. However, NPs internalised in the brain can induce cytotoxicity by oxidative stress, neuroinflammation, endoplasmic reticulum (ER) stress, disrupted signal pathways, and neurodegeneration in neurons [5,14]. Thus, the neurotoxicity of internalised NPs in the brain have been considered [15,16,17,18,19]. In addition, several issues have been noted in microglia exposed to nanoparticles, the primary immune cells in the brain, which migrate toward damaged regions and play a prominent role in neuroinflammation and neurodegeneration [20,21].

Microglia are the primary immune cells of the CNS. In response to injury and/or under pathological conditions, including treatment with NPs, microglia are activated, and they rapidly move toward the injured regions with an increase in the ATP levels [22]. The microglia role either neuroprotection by removing the tissue debris and pathogens or neuroinflammation by releasing inflammatory cytokines [14,23]. Their migration dynamics are related to immune responses and modulation in the brain [24,25]. Directed microglial migration, resulting in cell accumulation in damaged brain regions, is crucial to the immune responses mediated by microglial cells in the CNS [26]. Therefore, the activation of microglia and the effects of microglial migration mediated by NPs have been concerns in neuroscientific research.

There are limitations to the assessment of nanotoxicity with conventional analysis methods due to complex changes. Additionally, there is a lack of information about the neurotoxic effects of NPs. Therefore, systems toxicology, which involves the integration of traditional toxicology study and the examination of functional changes at multiple levels in molecules, is essential for the assessment of nanotoxicity [27]. In particular, omics analysis is a core method used in systems toxicology. However, a single-omics type cannot provide information related to interconnected molecular pathways in biological phenomena. In addition, integrated omics can be a more comprehensive and precise analysis than a single-omics analysis [28]. For example, integration of transcriptomics and metabolomics, namely metabotranscriptomics, demonstrated novel cellular responses and in-depth toxicological intracellular pathways associated with NPs of MNPs@SiO_2_(RITC) [29,30,31,32,33,34]. In addition, artificial intelligence (AI) approaches, such as machine learning, have been applied to classify the integration of omics data and predict nanotoxicity, which are massive and heterogeneous for generating a single network with integrated transcriptomics, proteomics, and metabolomics [35,36].

In the present study, we aimed to evaluate the mechanism of MNPs@SiO_2_(RITC)-induced toxicity on the migratory activity of microglia using single-omics and integrated triple-omics analysis. We also aimed to examine whether the downregulated migration activity was recovered by citrate.

## 2. Materials and Methods

### 2.1. MNPs@SiO_2_(RITC)

MNPs@SiO_2_(RITC) was purchased from BITERIALS (Seoul, South Korea). The size of the MNPs@SiO_2_(RITC) was determined to be 50 nm using TEM [33]. The *zeta* potential of MNPs@SiO_2_(RITC) was in the range of −40 to −30 mV [7,37]. X-ray diffraction analysis performed with a high-power X-ray diffractometer (Ultima III, Rigaku, Japan) was used to analyse the structure of MNPs@SiO_2_(RITC), which showed specific patterns of cobalt ferrite: (220) at 30°, (311) at 36°, (400) at 44°, (511) at 57°, and (440) at 64°. The broad peak observed between 20° and 40° indicated the presence of amorphous silica beads [33]. Rhodamine B-incorporated 50-nm silica NPs were purchased from CD Bioparticles (Shirley, NY, USA).

### 2.2. Cell Culture and Treatment with MNPs@SiO_2_(RITC)

BV2 cells were cultured as previously described, and the cell passages were used within 15 for maintaining morphological and functional properties [33,38,39]. Briefly, the cells were cultured in minimum essential medium (MEM, Gibco, Grand Island, NY, USA) supplemented with 10% foetal bovine serum (FBS, Gibco, Grand Island, NY, USA), 100 units/mL of penicillin, and 100 µg/mL of streptomycin (Gibco, Grand Island, NY, USA). The cells were incubated in a humidified chamber containing 5% CO_2_ at 37 °C. The effect of MNPs@SiO_2_(RITC) on the microglia was investigated after the treatment of BV2 cells with 0.01 and 0.1 µg/µL of MNPs@SiO_2_(RITC) for 12 h. Intracellular ROS generation and reduction in the intracellular ATP levels were analysed, and the trends were similar to a previous study (Appendix A) [35]. The uptake of MNPs@SiO_2_(RITC) almost reached a plateau upon treatment with 0.1 µg/µL MNPs@SiO_2_(RITC) (uptake ca 2.0 × 10^5^ particles per cell), and the viability of the BV2 cells was reduced by approximately 80% upon treatment with 0.01 µg/µL MNPs@SiO_2_(RITC) (uptake ca. 2.3 × 10^4^ particles per cell). The uptake efficiency was analysed by inductively coupled plasma mass spectrometry (ICP-MS) in a previous study [33]. Therefore, the optimal dose of MNPs@SiO_2_(RITC) for the treatment of microglia was determined as 0.01 to 0.1 µg/µL in the present study.

### 2.3. Migration Assay

The migration activity of BV2 cells was evaluated using a Transwell (Corning, Steuben, NY, USA) with an 8.0 μm pore size. The upper side of the insert was coated with 1:10 diluted with pH 8.0, 0.01 M Tris, 0.7% NaCl Matrigel (Corning, Steuben, NY, USA) for 2 h at 37 °C. The bottom chambers contained MEM supplemented with 10% FBS. MNPs@SiO_2_(RITC)-treated 10^4^ BV2 cells were overlaid onto the inserts and incubated for 12 h. The inserts were washed with phosphate-buffered saline (PBS), and the upper side of the inserts were cleaned with a cotton swab. The insert was fixed in Cytofix buffer (BD, San Jose, CA, USA) and stained with Hoechst 33342. After twice washing with PBS, images were acquired using an Axiovert 200M fluorescence microscope (Zeiss, Jena, Germany) at the 3D Immune System Imaging Core Facility of Ajou University. Migrated cells were counted in 5 random microscopic fields (378.27 mm^2^) using ImageJ software (NIH, Bethesda, MD, USA).

### 2.4. Cell Viability Assay

For the cell viability assay, a 3-(4,5-dimethylthiazol-2-yl)-5-(3-carboxymethoxyphenyl)-2-(4-sulfophenyl)-2H-tetrazolium (MTS) assay was performed using a CellTiter 96-cell proliferation assay kit (Promega, Madison, WI, USA) according to the manufacturer’s instructions. Briefly, aliquots containing 0.5 × 10^4^ cells were prepared in 96-well assay plates. The cells were treated with MNPs@SiO_2_(RITC) for 12 h. The cells were then washed with PBS to remove excess MNPs@SiO_2_(RITC). The MTS solution was added to each well of the 96-well plates, followed by incubation in an atmosphere containing 5% CO_2_ at 37 °C for 3 h.

### 2.5. Transcriptome Analysis

Transcriptomes of 0.01 and 0.1 µg/µL MNPs@SiO_2_(RITC)-treated BV2 cells were analysed, as previously described [33,35]. Briefly, total RNA was purified with a TruSeq Stranded Total RNA Library Prep Kit (Illumina, San Diego, CA, USA) after treatment with 0.01 and 0.1 µg/µL MNPs@SiO_2_(RITC) for 12 h. mRNA sequences with poly A tails were isolated with magnetic beads conjugated with oligo dT. The purified mRNAs were fragmented, and cDNA was obtained. The cDNAs were modified with poly-A, subjected to end-repair, and connected with sequencing adapters using the TruSeq RNA sample prep kit (Illumina, San Diego, CA, USA). cDNA fragments were purified with BluePippin (Sage Science, Beverly, MA, USA) and amplified by PCR. The library was sequenced using an Illumina HiSeq2500 sequencer (Illumina, San Diego, CA, USA). Differentially expressed genes were analysed as previously described [33,35].

### 2.6. Gene Ontology and Pathway Analysis

To analyse the gene ontology and pathway from omics data, ingenuity pathway analysis (IPA) bioinformatics software (Qiagen, Hilden, Germany) was used. A 1.5-fold change in gene and protein expression and a 1.2-fold change in the levels of the metabolites were used as cut-off values for estimation of significant differences in expression.

### 2.7. Quantitative Real-Time PCR (qPCR)

The expression levels of genes were analysed by qPCR. Gene-specific primer pairs (Appendix A) and an SsoAdvanced Universal SYBR Green Supermix real-time PCR kit (Bio-Rad, Hercules, CA, USA) were used for the qPCR. The reaction conditions were as follows: 95 °C for 5 min, followed by 50 cycles at 95 °C for 5 s and 60 °C for 30 s using a Rotor-Gene Q system (Qiagen, Hilden, Germany). The gene expression levels were analysed using melting curves generated by Rotor-Gene 1.7 software (Qiagen, Hilden, Germany). The PCR reactions were prepared independently in triplicate. The relative quantification of target gene expression was calculated using the 2^−ΔΔCt^ method.

### 2.8. Proteome Analysis

The proteomes of 0.01 and 0.1 µg/µL MNPs@SiO_2_(RITC)-treated BV2 cells were analysed as previously described [35]. Briefly, BV2 cells were lysed with RIPA buffer (Thermo Fisher Scientific, Cleveland, OH, USA) after treatment with 0.01 and 0.1 µg/µL MNPs@SiO_2_(RITC) for 12 h. The lysates were denatured with 8 M urea and reduced with 5 mM tris (2-carboxyethyl) phosphine hydrochloride at room temperature (RT) for 1 h. The samples were alkylated with 15 mM iodoacetamide in the dark at RT for 1 h. Proteins pooled from three biological replicate samples were digested with trypsin (Promega, USA) at 37 °C for 16 h. Each sample of 50 µg was labelled with tandem mass tag (TMT)-126 and -127 for the control, TMT-128 and -129 for the 0.01 µg/µL MNPs@SiO_2_(RITC)-treated group, and TMT-130 and -131 for the 0.1 µg/µL MNPs@SiO_2_(RITC)-treated group following the manufacturer’s protocol. The six TMT-labelled samples were combined and then divided using high-pH reversed-phase liquid chromatography (LC) into ten fractions. TMT-labelled peptides were analysed using an Easy nLC 1200 and Orbitrap Fusion Lumos mass spectrometer (Thermo Fisher Scientific, Cleveland, OH, USA) with high energy collision dissociation (HCD). Proteins were identified by two and more TMT-labelled peptide assignments at a false positive rate less than 0.01 and quantitatively compared by the average intensities of two reporter ions between the control and the two MNPs@SiO_2_(RITC)-treated groups.

### 2.9. Metabolic Profiling

Metabolic profiling of 0.01 and 0.1 µg/µL MNPs@SiO_2_(RITC)-treated BV2 cells was performed as previously described [32,35]. Briefly, ethoxycarbonyl (EOC)/methoxime (MO)/tert-butyldimethylsilyl (TBDMS) derivatives of 13 fatty acids (FAs), 20 amino acids (AAs), and 14 organic acids (OAs) were analysed using gas chromatography (GC)–MS/MS [32,35]. Cells were lysed by freezing/thawing. Quantification of each metabolite was conducted using a Shimadzu 2010 Plus gas chromatograph interfaced with a Shimadzu TQ 8040 triple quadruple mass spectrometer (Shimadzu, Kyoto, Japan).

### 2.10. Unsupervised Principal Component Analysis (PCA)

To trim the triple-omics factors (transcriptome, proteome, and metabolome) with relevance, we performed PCA and used machine learning algorithms as reported in a previous study [35]. The detected intensities in all omics analyses were normalised as Z-scores, and the Z-score of each triple-omics factor was adapted into a two-dimensional space, where the first dimension (PC1) and second dimension (PC2) were linear combinations of original values with a certain weight [40]. To find clusters in the PCA plot, we utilised an unsupervised algorithm implemented in *Scikit* [41]. In order to eliminate the outliers, the cut-off values were set as follows: −2.5 < PC1 < 3 and −1.3 < PC2 < 1.2.

### 2.11. Evaluation of Intracellular ATP Level

The ATP levels of 0.01 and 0.1 µg/µL MNPs@SiO_2_(RITC)-treated BV2 cells were evaluated as previously described [35]. Briefly, relative levels of intracellular ATP were evaluated using the CellTiter-Glo^®^ (Promega, Madison, WI, USA). The 0.01 and 0.1 μg/μL MNPs@SiO_2_(RITC)-treated BV2 cells were incubated at RT for 30 min, and luminescent reagent was added. The contents were mixed for 2 min and incubated for 10 min at RT. Luminescence signals were detected using a multi-mode microplate reader (SpectraMax^®^ iD3; Molecular Devices, San Jose, CA, USA).

### 2.12. Flow Cytometry for Evaluation of Relative Intracellular NPs Amount

BV2 cells were treated with 0.01 and 0.1 µg/µL of MNPs@SiO_2_(RITC) and silica NPs for 12 h and then treated citric acid for 12 h. The cells were trypsinised and washed twice with PBS and fixed in Cytofix buffer (BD, San Jose, CA, USA) at 4 °C for 20 min. The cells were washed twice with PBS. The labelled cells were analysed using a flow cytometer (BD FACS Aria II™, BD, San Jose, CA, USA) at the 3D Immune System Imaging Core Facility of Ajou University.

### 2.13. Immunocytochemistry

The cells were seeded on coverslips and treated with 0.01 and 0.1 µg/µL of MNPs@SiO_2_(RITC) for 12 h. The cells were then fixed in Cytofix buffer (BD Biosciences, San Jose, CA, USA). To reduce non-specific binding, the coverslips were blocked with PBS containing 2% BSA and 0.1% Triton-X100 (Sigma-Aldrich, St Louis, MO, USA). The cells were incubated with Alexa Fluor 488-conjugated phalloidin (Molecular probe, USA 1:200), diluted in blocking buffer, for 1 h at RT. The labelled cells were washed thrice with PBS containing 0.1% Triton-X100 and incubated with PBS containing 10 µg/mL Hoechst 33342 for 10 min at RT for nuclear labelling. After washing with PBS, the coverslips were mounted onto slides using ProLong Gold Antifade mounting medium (Molecular Probes, USA). Fluorescence images were acquired by confocal laser scanning microscopy (LSM710, Carl Zeiss Microscopy GmbH, Jena, Germany) at the 3D Immune System Imaging Core Facility of Ajou University. The excitation wavelengths for Alexa Fluor 488, Hoechst 33342, and MNPs@SiO_2_(RITC) were 488, 405, and 530 nm, respectively.

### 2.14. Statistical Analysis

Data were analysed using one-way analysis of variance (ANOVA) using IBM-SPSS software (IBM, USA). Bonferroni’s multiple comparison test was used as a *post hoc* test. *p*-values lower than 0.05 were considered as significant changes.

## 3. Results

### 3.1. Reduction in the Migratory Activity in MNPs@SiO_2_(RITC)-Treated BV2 Cells

Changes in the migratory activity of MNPs@SiO_2_(RITC)-treated BV2 immortalised murine microglial cells were analysed using Transwell inserts with 8 μm pores. The BV2 cells were treated with 0.01 or 0.1 μg/μL MNPs@SiO_2_(RITC) for 12 h and transferred to the insert, followed by invasion induced by chemotaxis (Figure 1A). No significant changes were observed in the viability of MNPs@SiO_2_(RITC)-treated BV2 cells (Figure 1B). The number of migrating MNPs@SiO_2_(RITC)-treated BV2 cells was significantly decreased in a dose-dependent manner (Figure 1C,D). We also analysed intracellular ROS generation and reduction in the intracellular ATP levels, and the trends were similar to those observed in a previous study (Appendix A) [35].

### 3.2. Transcriptomic Analysis of MNPs@SiO_2_(RITC)-Treated BV2 Cells

We previously analysed changes in the transcriptome of MNPs@SiO_2_(RITC)-treated BV2 cells using RNA-seq, and we identified 4760 differentially expressed genes (fold change cut-off ± 1.5) from a total of 41,214 analysed genes [33,35]. Moreover, gene ontologies in canonical pathways and biological functions were analysed (Appendix A), and ‘cellular movement’ was ranked within the top 15 biological functions. The expression of 48 genes associated with the cellular movement was significantly altered in 0.1 μg/μL MNPs@SiO_2_(RITC)-treated BV2 cells (Figure 2A). A transcriptomic network was generated with 48 genes using IPA (Appendix A), and we found genes related to ROS, ATP concentration, and cell migration (Figure 2B and Appendix A). Moreover, in silico prediction of the network showed upregulation of ROS, downregulation of ATP concentration, and downregulation of cell migration. Among the genes in the network, expression levels of formyl peptide receptor 1 (*Fpr1*), protein kinase C, beta (*Prkcb*), serine (or cysteine) SH2 domain protein 2A (*Sh2d2a*), and microtubule-associated protein tau (*Mapt*) were validated by qPCR (Figure 2C), and the tendency of changes was similar between the results of RNA-seq and those of qPCR analysis.

### 3.3. Proteomic Analysis of MNPs@SiO_2_(RITC)-Treated BV2 Cells

We previously analysed changes in the proteome of MNPs@SiO_2_(RITC)-treated BV2 cells using LC–MS/MS and identified 482 differentially expressed proteins (fold change cutoff ± 1.5) from a total of 5918 analysed proteins [35]. Moreover, the gene ontologies of canonical pathways and biological functions were analysed (Appendix A), and ‘cellular movement’ was ranked within the top 15 biological functions. The expression of 59 proteins associated with the cellular movement was significantly altered in 0.1 μg/μL MNPs@SiO_2_(RITC)-treated BV2 cells (Figure 3A). A proteomic network was generated with 59 proteins using IPA (Appendix A), and we found that the proteins were related to ROS, ATP concentration, and cell migration (Figure 3B and Appendix A). Moreover, in silico prediction of the network showed upregulation of ROS, downregulation of ATP concentration, and downregulation of cell migration. Among the proteins in the network, the relative abundance levels of myosin heavy chain 9 (Myh9), membrane-bound O-acyltransferase domain containing 7 (Mboat7), POU class 2 homeobox 1 (Pou2f1), and the cluster of differentiation 80 (Cd80) tended to be up-or downregulated, similar to the results observed with the network analysis (Figure 3C).

### 3.4. Metabolomic Analysis of MNPs@SiO_2_(RITC)-Treated BV2 Cells

Metabolic profiling of 13 FAs, 20 AAs, and 14 OAs was previously analysed using EOC/MO/TBDMS derivatisation and GC–MS/MS [32,35]. Among the metabolites with altered levels (fold change cutoff ± 1.2), reduced levels of three FAs, two AAs, and seven OAs, and increased levels of two FAs, two AAs, and one OA were found in 0.1 μg/μL MNPs@SiO_2_(RITC)-treated BV2 cells [35]. Moreover, metabolic profile-related canonical pathways and biological functions were analysed (Appendix A). Among the altered levels of metabolites, we found that nine metabolites were highly related to biological functions, as deduced by transcriptomics and proteomics analysis (Figure 4A), and we generated a metabolic network using IPA (Figure 4B and Appendix A). Moreover, in silico prediction of the network also showed upregulation of ROS, downregulation of ATP concentration, and downregulation of cell migration. Representative selected-ion monitoring chromatograms of succinic acid, citric acid, arachidonic acid, and malonic acid in the metabolomic network are shown in Figure 4C.

### 3.5. Integrated Omics Analysis of Biological Functions in MNPs@SiO_2_(RITC)-Treated BV2 Cells

Each omics network was combined into a single triple-omics network to compensate for the potential weaknesses of each omics approach. In silico predictions for ROS, ATP concentration, and cell migration were consistent with single-omics network predictions. To reduce the complexity of multi-omics networks, we eliminated the controversial relationship between molecules and biological function. The network consisted of 31 genes, 45 proteins, and nine metabolites (Figure 5A and Appendix A). The trimmed network showed strong relationships among genes, proteins, metabolites, and biological functions.

The expression levels of genes, proteins, and metabolites (factors) in integrated triple-omics were clustered using machine learning-based unsupervised analysis. The overall expression levels of factors are shown in Figure 5B. Based on the data distribution, we identified two major clusters. In addition, the factors congregated within two major dominant clusters, indicating strong relationships. To further evaluate the relationships among these triple-omics factors, we trimmed the network using the two major clusters in Figure 5B (indicated by filled circles) by eliminating outlier factors for clusters (Appendix A). We performed a prediction of the trimmed network (Figure 5C, Appendix A), and the trimmed integrated networks showed strong associations between the factors and biological functions.

### 3.6. Alleviation Effect of Citrate on the Reduction of Migratory Activity in MNPs@SiO_2_(RITC)-Treated BV2 Cells

Among the triple-omics networks, reduction in OA levels, which elucidate suppression of the tricarboxylic acid (TCA) cycle, was dominantly observed and highly related to other genes and proteins. We hypothesised that supplementation with OAs could be employed as a strategy for the alleviation of MNPs@SiO_2_(RITC)-induced migratory activity reduction. Among the OAs, citrate is a candidate for OA supplementation according to the expression of citrate transporters in the plasma membrane and mitochondrial membrane for proper delivery, and the citrate treatment concentration of 1 mM was determined as optimal and maximum concentration to alleviate MNPs@SiO_2_(RITC)-induced toxicity in a previous study [42]. The migratory activity in MNPs@SiO_2_(RITC)-treated BV2 cells was analysed in the presence of citrate (Figure 6A). The migratory activity was decreased in an MNPs@SiO_2_(RITC)-dose-dependent manner, and this decrease was alleviated in citrate co-treated BV2 cells (Figure 6B). No significant changes were observed in the cell viability in MNPs@SiO_2_(RITC)-treated BV2 cells or citrate co-treated BV2 cells (Figure 6C). However, intracellular ATP levels were reduced in MNPs@SiO_2_(RITC)-treated cells, and the intracellular ATP reduction was alleviated by citrate co-treatment (Figure 6D).

### 3.7. Increase in Nanoparticle Exocytosis in BV2 Cells Induced by Citrate Treatment

Increments of intracellular ATP levels are highly related to the active transport of molecules, including exocytosis [43]. Interestingly, we found that the triple-omics single network trimmed using a machine learning algorithm is significantly associated with exocytosis, and prediction of the network demonstrated a suppression in exocytosis (Figure 7A and Appendix A). We examined changes in the nanoparticle exocytosis status in the presence of citrate induced by alleviating the reduction in intracellular ATP levels. BV2 cells were treated with MNPs@SiO_2_(RITC) and rhodamine B-incorporated silica NPs for 12 h and then treated with citrate for 12 h (Figure 7B,C). Increased levels of fluorescence due to an increase in the internalised nanoparticles were analysed using FACS. The fluorescence levels were increased in an NP-dose-dependent manner, and the fluorescence levels were reduced by ~10% in the NP- and citrate-treated groups (Figure 7D,E).

## 4. Discussion

The results of this study performed using the systems toxicological approach demonstrated that the migratory activity of MNPs@SiO_2_(RITC)-treated BV2 cells was reduced. In addition, we found that the migratory activity of MNPs@SiO_2_(RITC)-treated BV2 cells was recovered by treatment with citrate due to an increase in the ATP levels and the exocytosis effect. Therefore, optimal concentrations of NPs and co-treatment of drugs with reduced nanotoxicity are recommended strategies for NP application in the CNS.

The migratory activity of microglia is important for protective effects associated with immune functions in the CNS because activated microglia move toward the injured lesion area and secrete inflammatory cytokines and neurotrophic factors and induce phagocytosis of foreign substances, including NPs or cell debris [24,44]. Typically, cell migration is modulated by the ATP/ADP ratio [45] and lipid peroxidation by lipid peroxyl radicals [46]. In our previous study, 0.1 μg/μL MNPs@SiO_2_(RITC)-treated cells showed decreased intracellular ATP levels and increased lipid peroxidation in the membrane due to the production of ROS, which can reduce membrane fluidity and traction force [47]. Hence, we speculated that MNPs@SiO_2_(RITC) might reduce the migratory activity of microglia by reducing intracellular ATP levels, synthesis of ROS, and lipid peroxidation in the membrane.

Citrate has been purposed in a previous study as an alleviation agent for nanomaterial-induced toxicity by the following effects: (i) nutritional supplementation for TCA cycle intermediate; (ii) chelating intracellular calcium ion and toxic NP-derived metal ion [35]. In this study, we hypothesized that ATP levels increase upon treatment with citrate in MNPs@SiO_2_(RITC)-treated BV2 cells with an increase in the migratory activity, as citrate is converted into isocitrate via cis-aconitate by aconitase for the production of ATP in mitochondria [48]. In addition, citrate has been used as an NP stabilizer by a capping agent and storage formulation [49,50,51]. Thus, conditionally releasable citrate capping or utilization of citrate as an external agent might be a strategy for alleviation of NP-induced toxicity.

Formation of focal adhesions and reorganization of the cytoskeleton are important biological functions for regulation of migratory activity of microglia [52,53]. We found that the biological functions are associated with the triple-omics single network trimmed using a machine learning algorithm (Appendix A), and the formation of focal adhesions and reorganization of the cytoskeleton were predicted as suppressed in MNPs@SiO_2_(RITC)-treated BV2 cells (Appendix A). Moreover, a previous study suggested that the migratory activity of microglia is regulated not only through gene expressions but also can be regulated by activation/inactivation of intracellular signal molecules, including those regulating focal adhesion formation and actin re-organization [52,53]. The detailed activation/inactivation of intracellular signal molecules might be analysed with phosphoproteome analysis using LC–MS/MS in future studies.

Previous studies demonstrated that the biological effects of MNPs@SiO_2_(RITC) and 50 nm silica NPs, which are similar to MNPs@SiO_2_(RITC) except for the lack of a cobalt ferrite core, are identical in many mammalian cells [29,30,31,37]. Thus, we hypothesized that reduced migration activity of MNPs@SiO_2_(RITC)-treated BV2 cells can be attributed to the silica shell rather than the core component, namely cobalt ferrite. In addition, the migration pattern of the BV2 cell line was very similar to the pattern of primary microglia originated from mouse, rat, and human [35,54,55]. However, the effect of MNPs@SiO2(RITC) majorly depends on physical interaction between cells and NPs, and the reduction of migration activity is expected to be much different between the BV2 cell line and each primary microglia. Further studies are needed concerning the differences of migration patterns between BV2 cell line and primary microglia.

An increment in intracellular ATP levels is significantly associated with the active transport of molecules, including exocytosis [43]. When citrate was added to MNPs@SiO_2_(RITC)-treated BV2 cells, the exocytosis of MNPs@SiO_2_(RITC) increased, which can be attributed to ATP acting as an energy source for NP exocytosis. The physicochemical properties of NPs, such as size, shape, and surface properties, can affect exocytosis [56,57]. Thus, further studies are needed to investigate the physicochemical properties of NPs and their exocytosis efficiency with or without citrate. In addition, citrate is a calcium chelator [58], and NPs increase cytosolic Ca^2+^ concentrations with Ca^2+^ pumps on the plasma membrane and the mitochondrial membrane, inducing Ca^2+^ toxicity in NP-treated cells [59]. Although we did not obtain direct evidence of calcium chelation induced by citrate, we hypothesized that citrate may act as a chelating agent of calcium produced by nanoparticles to reduce nanotoxicity. So far, due to the usefulness of NPs, compared to the studies examining endocytosis of NPs, fewer studies examining NP exocytosis have been published [43,60,61]. Therefore, further studies are needed to investigate the chelating effects of citrate in NP-treated cells. Moreover, we examined the NPs’ exocytosis by the treatment of citrate using FACS in this study, and the effect was too subtle for clear conclusion. Thus, detailed NP exocytosis analysis is required using quantitative analytic methods such as ICP-MS in a future study.

The omics data were processed using the PCA and K-means clustering algorithm by reducing the dimensionality of the data. Processing is used for clustering or removing outliers using cut-off values. Moreover, in the processing of data using machine learning algorithms, approximately 20% of the data were trimmed to reduce biases in the analysis and to obtain stronger relationships of data [35]. This approach will be useful in the analysis and data processing of integrated omics data, including transcriptome, proteome, metabolome, and small RNA.

In conclusion, the present study suggests that exposure to NPs might decrease the migratory activity of microglia, which can be recovered by citrate, based on a combination of conventional biomolecular methods, RNA-seq-based transcriptomics, and metabolomics analyses used to evaluate the activation of microglia treated with MNPs@SiO_2_(RITC). Although an in vivo experiment will be a direct way to show the effects of MNPs@SiO_2_(RITC), the data obtained from BV2 cells, including integrated omics analysis, can be used to extrapolate the phenomena in vivo. Our findings highlight the importance of minimization of dosages in NP applications of brain-targeting diagnostics and drug delivery to reduce possible adverse effects. In addition, this study suggests the co-treatment with agents that reduce nanotoxic effects on microglia when NPs are used as therapeutic and diagnostic agents for brain diseases.

## Figures and Tables

**Figure 1 cells-11-02393-f001:**
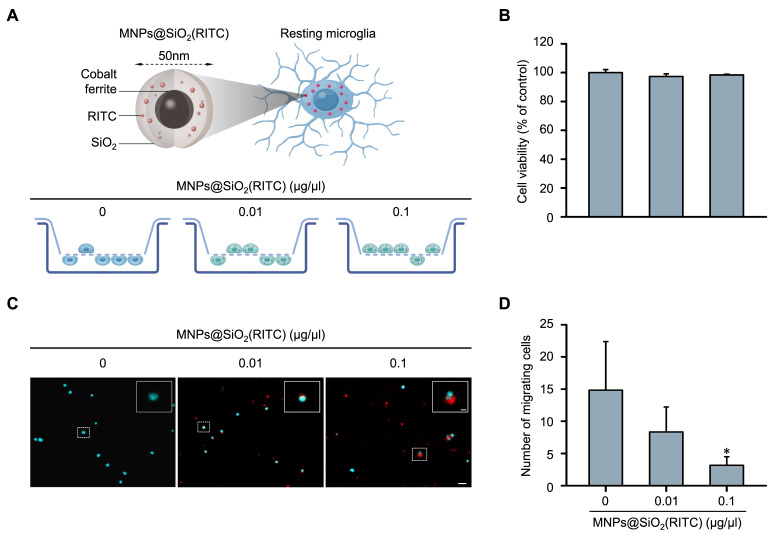
Reduction in migratory activity of MNPs@SiO_2_(RITC)-treated BV2 cells. (**A**) Schematic representation of the invasion assay performed with MNPs@SiO_2_(RITC)-treated BV2 cells. Non-treated control BV2 cells, 0.01 µg/µL MNPs@SiO_2_(RITC)-treated BV2 cells, and 0.1 µg/µL MNPs@SiO_2_(RITC)-treated BV2 cells were used. (**B**) Viability of MNPs@SiO_2_(RITC)-treated BV2 cells for 12 h. (**C**) Invasion assay of MNPs@SiO_2_(RITC)-treated BV2 cells. The nuclei of the cells were stained using Hoechst 33342, and the fluorescence was expressed as cyan. Fluorescence of MNPs@SiO_2_(RITC) distribution was expressed as red. The number of nuclei was counted in five randomly selected fields. Scale bar = 50 μm. Scale bar of cropped image = 5 μm. (**D**) Number of migrating cells. Data represent means ± standard deviation derived from three independent experiments. * *p* < 0.05 vs. control.

**Figure 2 cells-11-02393-f002:**
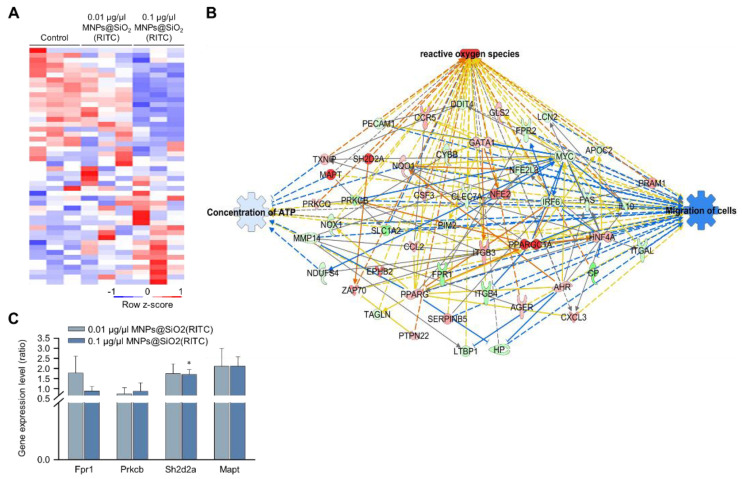
Transcriptomic analysis in MNPs@SiO_2_(RITC)-treated BV2 cells. (**A**) Heatmap of 48 genes related to cellular movement in the transcriptome of MNPs@SiO_2_(RITC)-treated BV2 cells observed with RNA-seq analysis. (**B**) Functional analysis of the transcriptomic network with prediction using IPA in 0.1 μg/μL MNPs@SiO_2_(RITC)-treated BV2 cells. The analysis involved a fold change cut-off value of ±1.5. Red and green nodes indicate genes whose expressions were upregulated and downregulated, respectively, compared to those of the control. Orange and blue arrows indicate the prediction of activation and inhibition, respectively. Details of shape and colour, which are originated from IPA. (**C**) qPCR analysis was performed to validate gene expression in each group. *Gapdh* was used as an internal control. Data represent means ± standard deviations of three independent experiments. * *p* < 0.05 vs. control.

**Figure 3 cells-11-02393-f003:**
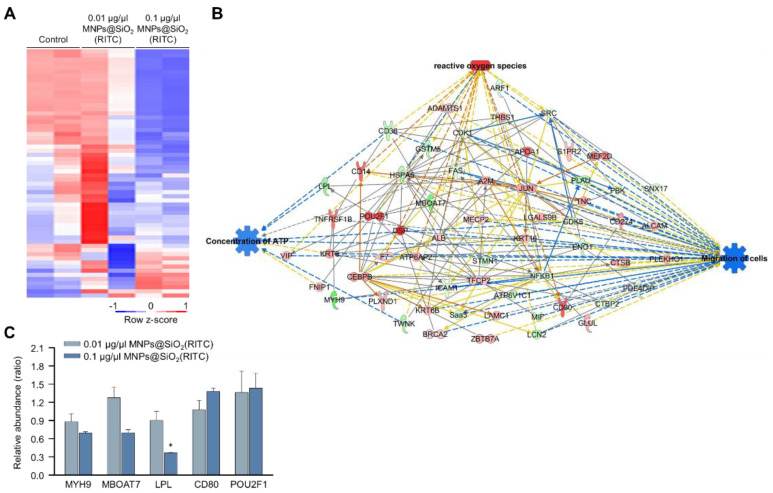
Proteomic analysis of MNPs@SiO_2_(RITC)-treated BV2 cells. (**A**) Heatmap of 60 proteins related to the cellular movement in the proteome of MNPs@SiO_2_(RITC)-treated BV2 cells from LC–MS/MS analysis. (**B**) Functional analysis of the proteomic network in 0.1 μg/μL MNPs@SiO_2_(RITC)-treated BV2 cells with prediction using IPA. The analysis involved a fold change cut-off value of ±1.5. Red and green nodes indicate genes whose expressions were upregulated and downregulated, respectively, compared to those of the control. Orange and blue arrows indicate the prediction of activation and inhibition, respectively. Details of shape and colour, which are originated from IPA. (**C**) Relative abundance levels of Myh9, Mboat7, Lpl, Pou2f1, and Cd80 according to LC–MS/MS analysis. Data represent the means ± standard deviation of the two TMT ratios. * *p* < 0.05 vs. control.

**Figure 4 cells-11-02393-f004:**
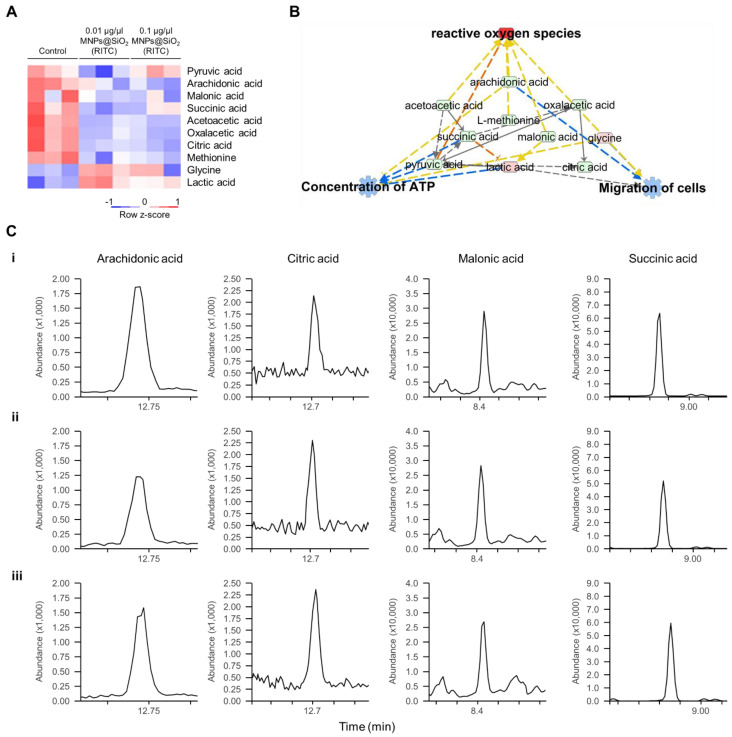
Metabolic profiling of MNPs@SiO_2_(RITC)-treated BV2 cells. (**A**) Heatmap of nine metabolites related to the cellular movement in the metabolome of MNPs@SiO_2_(RITC)-treated BV2 cells as observed with GC–MS/MS analyses. (**B**) Functional analysis of the metabolomic network in 0.1 μg/μL MNPs@SiO_2_(RITC)-treated BV2 cells with prediction using IPA. The analysis involved a fold change cut-off value of ±1.2. Red and green nodes indicate genes whose expressions were upregulated and downregulated, respectively, compared to those of the control. Orange and blue arrows indicate the prediction of activation and inhibition, respectively. Details of the shape and colour, which are originated from IPA. (**C**) Representative selected-ion monitoring chromatograms of arachidonic acid, citric acid, malonic acid, and succinic acid for (**i**) control, (**ii**) 0.01, and (**iii**) 0.1 µg/µL MNPs@SiO_2_(RITC)-treated BV2 cells.

**Figure 5 cells-11-02393-f005:**
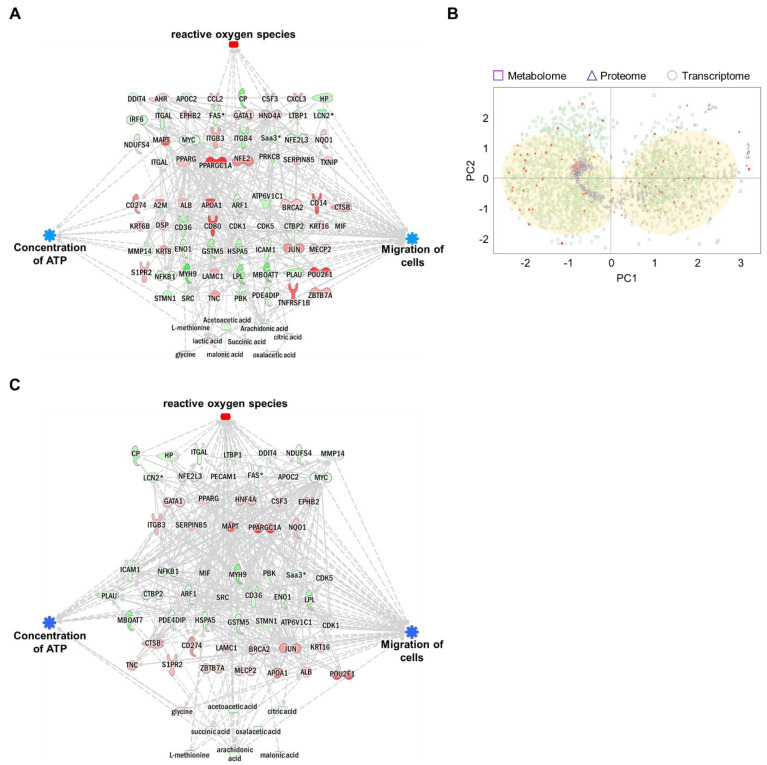
Construction of an integrated triple-omics network in MNPs@SiO_2_(RITC)-treated BV2 cells. (**A**) The merged and trimmed triple-omics network of 0.1 µg/µL MNPs@SiO_2_(RITC)-treated BV2 cells with the prediction. Top group: transcriptome, middle group: proteome, bottom group: metabolome. Orange and blue areas indicate prediction as activation and inhibition, respectively. Downregulated Fas, Lcn2, Saa3, and Mmp14 expressions were shared between the transcriptome and proteome. Symbols are described in the legend of Appendix A. (**B**) PCA for triple-omics factors. Filled spots indicate items included in the simply merged triple-omics network. The translucent yellow circle indicates the location of the major dominant cluster. (**C**) The trimmed network using the machine learning algorithm and integrated prediction network for four categories of biological functions of omics data derived from 0.1 µg/µL MNPs@SiO_2_(RITC)-treated BV2 cells.

**Figure 6 cells-11-02393-f006:**
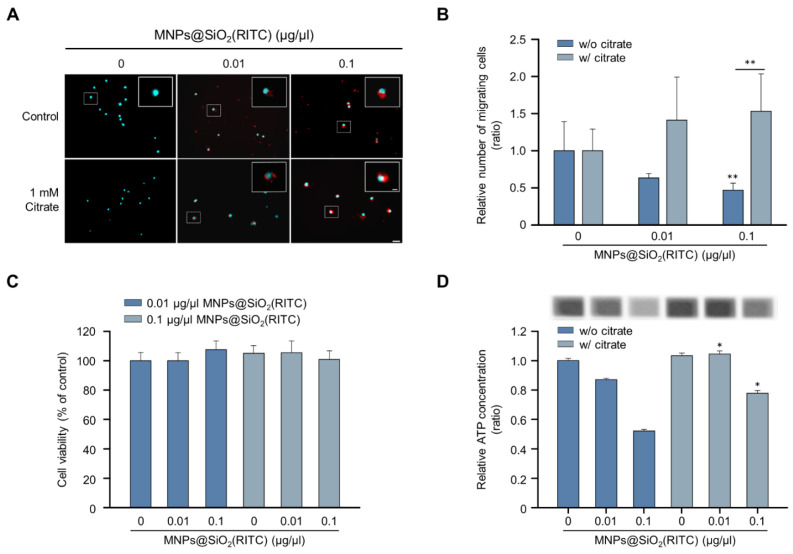
Recovered migratory activity of MNPs@SiO_2_(RITC)-treated BV2 cells by citrate treatment. (**A**) Transwell assay of MNPs@SiO_2_(RITC) and citrate co-treated BV2 cells. The nuclei of the cells were stained using Hoechst 33342, and the fluorescence was expressed as cyan. Fluorescence of MNPs@SiO_2_(RITC) distribution was expressed as red. The cells were counted under a fluorescence microscope in five randomly selected fields. The number of nuclei was counted in five randomly selected fields. Scale bar = 50 μm. Scale bar of cropped image = 10 μm. (**B**) The number of migrated cells. Data represent means ± standard deviation of three independent experiments. * *p* < 0.05 vs. control. ** *p* < 0.01 vs. control. (**C**) Viability of MNPs@SiO_2_(RITC)-treated BV2 cells in the presence of 1 mM citrate for 12 h. (**D**) Intracellular ATP levels in MNPs@SiO_2_(RITC)-treated BV2 in the presence of 1 mM citrate for 12 h. Data represent means ± standard deviation of three independent experiments. * *p* < 0.05 vs. control.

**Figure 7 cells-11-02393-f007:**
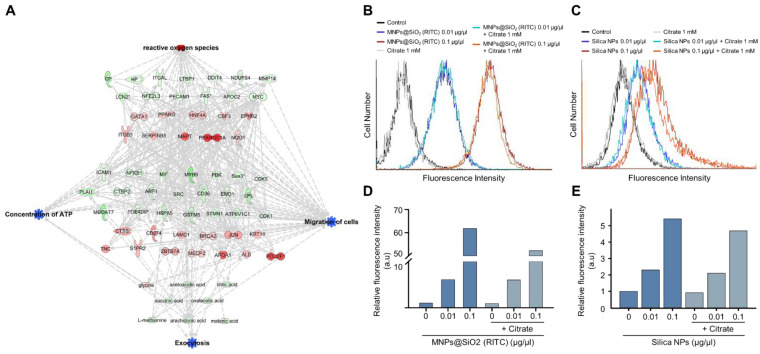
Induction of exocytosis in MNPs@SiO_2_(RITC)- and silica NP-treated BV2 cells by citrate treatment. (**A**) Trimmed using machine learning algorithm triple-omics network with the prediction of 0.1 µg/µL MNPs@SiO_2_(RITC)-treated BV2 cells including exocytosis. Top group: transcriptome, middle group: proteome, bottom group: metabolome. Symbols are described in the legend of Appendix A. Orange and blue areas indicate the prediction as activation and inhibition, respectively. FACS analysis for evaluation of fluorescence of (**B**) MNPs@SiO_2_(RITC) and (**C**) silica NP-treated BV2 cells in the presence of 1 mM citrate. Relative fluorescence intensity of (**D**) MNPs@SiO_2_(RITC) and (**E**) silica NPs in BV2 cells. a.u.: arbitrary unit.

## Data Availability

The data supporting the findings of this study are available from the corresponding author upon reasonable request. Transcriptome sequencing and quantification data are available in the GEO database under the following accession number: GSE154250 [33,35]. Proteome and quantification data are available in PRIDE with the following accession number: PXD020225 [35].

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
