# Peer review of "Reduction in the Migration Activity of Microglia Treated with Silica-Coated Magnetic Nanoparticles and their Recovery Using Citrate"

_cells, 2022, doi:10.3390/cells11152393_

Round 1

Reviewer 1 Report

Nanoparticles (NPs) are expected to be a useful method for delivering drugs into the brain. However, its neurotoxicity has not been investigated. In the previous study, the authors showed that silica-coated-magnetic-NPs were toxic to microglia.  In addition, as was predicted by omics analysis, citric acid recovered the impairment of microglia.  The present study showed that silica-coated-magnetic-NPs reduced migration of microglia, and that citric acid improved it.  To examine effects of NPs on brain cells is of significance. So, the present study is meaningful. Considering the following points, the present conclusion will be more convincing.

1.    In the previous report [ref.32], primary culture microglia were used in addition to BV2, but not used in the present study.  Because established cell lines, including BV2, often lose the properties of the original cells, it should be confirmed whether similar reductions of migration activity by NPs are obtained in primary cultured microglia.

2.    It is of great interest to examine whether the reduction of microglial migration by the NPs can be observed in vivo.  The previous report showed microglia accumulated intraperitoneal administrated NPs.  Exanimating whether microglial migration to injured brain region is reduced by intraperitoneal administrated NPs would make the present conclusion more convincing.

3.    The authors think that reduction of microglial migration by NPs is caused by genetic changes associated with cell damage, and analyze the mechanism by omics analysis. While this methodology is not denied, microglial migration is regulated not only through gene expressions but also by activation/inactivation of intracellular signal molecules, including those regulating focal adhesion formation and actin re-organization (Mol Cells. 2017, 40:163-168.).   Alterations of these signals which are not mediated by gene expressions should also be examined.

4.    Regarding Figure 6D, the method for measuring intracellular ATP is presented only in supplementary information of the previous report, but not in this article. Please include it in this article.  Also, regarding measurement of exocytosis, the method should be specified in section 2.11.

5.    The authors conclude that recovery of BV2 migration by citrate is caused by stimulation of NP exocytosis.  This is predicted from results of the omics analysis. However, the data in Figure 7B-E show only a small increase in NP release by citrate, and are not confirmed by statistical analysis.  So, it is hard to conclude that citrate stimulates NP exocytosis based on these data. The possibility that citrate promotes NP exocytosis should be discussed only when statistically significant citrate-induced NP releases is obtained, after the same experiments are repeated.

Author Response

Answers to the comments of Reviewer #1

Nanoparticles (NPs) are expected to be a useful method for delivering drugs into the brain. However, its neurotoxicity has not been investigated. In the previous study, the authors showed that silica-coated-magnetic-NPs were toxic to microglia. In addition, as was predicted by omics analysis, citric acid recovered the impairment of microglia. The present study showed that silica-coated-magnetic-NPs reduced migration of microglia, and that citric acid improved it. To examine effects of NPs on brain cells is of significance. So, the present study is meaningful. Considering the following points, the present conclusion will be more convincing.

  1. In the previous report [ref.32], primary culture microglia were used in addition to BV2, but not used in the present study. Because established cell lines, including BV2, often lose the properties of the original cells, it should be confirmed whether similar reductions of migration activity by NPs are obtained in primary cultured microglia.

à We agree with your concern. However, we did not find the morphological and functional alterations according to the properties of the original cells, because all experiments were performed with early passage cells and the cells were discarded after 15 passages. In addition, the migration pattern of BV2 cell line was very similar with the pattern of primary microglia originated from mouse, rat, and human (REF 35, 54, 55). However, the effect of MNPs@SiO2(RITC) majorly depends on physical interaction between cells and NPs and the reduction of migration activity is expected to be much different between BV2 cell line and each primary cultured microglia. We added references and this point to

Line 100, Page 3

2.2. Cell culture and treatment with MNPs@SiO2(RITC)

BV2 cells were cultured as previously described and the cell passage were used within 15 for maintaining morphological and functional properties [33,38,39]. Briefly, the cells were cultured in minimum essential medium (MEM, Gibco, Grand Island, NY, USA) supplemented with 10% foetal bovine serum (FBS, Gibco, Grand Island, NY, USA), 100 units/ml of penicillin, and 100 µg/ml of streptomycin (Gibco, Grand Island, NY, USA). The cells were incubated in a humidified chamber containing 5% CO2 at 37°C. The effect of MNPs@SiO2(RITC) on the microglia was investigated after the treatment of BV2 cells with 0.01 and 0.1 µg/µl of MNPs@SiO2(RITC) for 12 h. Intracellular ROS generation and reduction in the intracellular ATP levels were analysed and the trends were similar with previous study (Figure S1) [35]. The uptake of MNPs@SiO2(RITC) almost reached a plateau upon treatment with 0.1 µg/µl MNPs@SiO2(RITC) (uptake ca 2.0 × 105 particles per cell), and the viability of the BV2 cells was reduced by approximately 80% upon treatment with 0.01 µg/µl MNPs@SiO2(RITC) (uptake ca. 2.3 × 104 particles per cell). The uptake efficiency was analysed by inductively coupled plasma mass spectrometry (ICP-MS) in previous study [33]. Therefore, the optimal dose of MNPs@SiO2(RITC) for the treatment of microglia was determined as 0.01 to 0.1 µg/µl in the present study.

Additional References

  1. Rangaraju, S.; Raza, S.A.; Li, N.X.; Betarbet, R.; Dammer, E.B.; Duong, D.; Lah, J.J.; Seyfried, N.T.; Levey, A.I. Differential Phagocytic Properties of CD45(low) Microglia and CD45(high) Brain Mononuclear Phagocytes-Activation and Age-Related Effects. Front Immunol. 2018, 9, 405.
  2. Stansley, B.; Post, J.; Hensley, K. A comparative review of cell culture systems for the study of microglial biology in Alzheimer's disease. J Neuroinflammation. 2012, 9, 115.

Line 473, Page 13

Previous studies demonstrated that the biological effects of MNPs@SiO2(RITC) and 50 nm silica NPs, which are similar to MNPs@SiO2(RITC) except for the lack of a cobalt ferrite core, are identical in many mammalian cells [29-31,37]. Thus, we hypothesized that reduced migration activity of MNPs@SiO2(RITC)-treated BV2 cells can be attributed to the silica shell rather than the core components, cobalt ferrite. In addition, the migration pattern of BV2 cell line was very similar with the pattern of primary microglia originated from mouse, rat, and human [35,54,55]. However, the effect of MNPs@SiO2(RITC) majorly depends on physical interaction between cells and NPs and the reduction of migration activity is expected to be much different between BV2 cell line and each primary microglia. Further studies are needed concerning the difference of migration pattern between BV2 cell line and primary microglia.

Additional References

  1. Anton, R.; Ghenghea, M.; Ristoiu, V.; Gattlen, C.; Suter, M.R.; Cojocaru, P.A.; Popa-Wagner, A.; Catalin, B.; Deftu, A.F. Potassium Channels Kv1.3 and Kir2.1 But Not Kv1.5 Contribute to BV2 Cell Line and Primary Microglial Migration. Int J Mol Sci. 2021, 22, 2081.
  2. Flora, G.K.; Anderton, R.S.; Meloni, B.P.; Guillemin, G.J.; Knuckey, N.W.; MacDougall, G.; Matthews, V.; Boulos, S. Microglia are both a source and target of extracellular cyclophilin A. Heliyon. 2019, 5, e02390.

  1. It is of great interest to examine whether the reduction of microglial migration by the NPs can be observed in vivo. The previous report showed microglia accumulated intraperitoneal administrated NPs. Exanimating whether microglial migration to injured brain region is reduced by intraperitoneal administrated NPs would make the present conclusion more convincing.

à As you recommended, exanimating whether microglial migration to injured brain region will be helpful for making convince conclusion. Although in vivo experiment will be direct way for showing the effect of MNPs@SiO2(RITC), the data obtained from BV2 cells, including integrated omics analysis, was used to extrapolate the phenomena in in vivo. We added these points to the revised manuscript as follows

Line 508, Page 14

In conclusion, the present study suggests that exposure to NPs might decrease the migratory activity of microglia, which can be recovered by citrate, based on a combination of conventional biomolecular methods, RNA-seq-based transcriptomics, and metabolomics analyses used to evaluate the activation of microglia treated with MNPs@SiO2(RITC). Although in vivo experiment will be direct way for showing the effect of MNPs@SiO2(RITC), the data obtained from BV2 cells, including integrated omics analysis, can be used to extrapolate the phenomena in in vivo. Our findings highlight the importance for minimization of dosages in NPs application of brain-targeting diagnostics and drug delivery to reduce possible adverse effects. In addition, this study suggests that co-treatment with agents that reduce nanotoxic effects on microglia when NPs are used as therapeutic and diagnostic agents for brain diseases.

  1. The authors think that reduction of microglial migration by NPs is caused by genetic changes associated with cell damage, and analyze the mechanism by omics analysis. While this methodology is not denied, microglial migration is regulated not only through gene expressions but also by activation/inactivation of intracellular signal molecules, including those regulating focal adhesion formation and actin re-organization (Mol Cells. 2017, 40:163-168.). Alterations of these signals which are not mediated by gene expressions should also be examined.

à We agree with your opinion. Although the activation/inactivation of intracellular signal molecules also are important regulator of microglial migration, we found additional relationships with the regulating focal adhesion formation and actin re-organization in our additional analysis using triple-omics single network trimmed using a machine learning algorithm. The detailed activation/inactivation of intracellular signal molecules might be analyzed with phosphoproteome analysis using LC-MS/MS as a further study. We added these points to the revised manuscript as follows

Line 458, Page 13

Formation of focal adhesions and reorganization of cytoskeleton are important biological functions for regulation of migratory activity of microglia [52,53]. We found that the biological functions are associated with the triple-omics single network trimmed using a machine learning algorithm (Figures S21, S22), and the formation of focal adhesions and reorganization of cytoskeleton were predicted as suppressed in MNPs@SiO2(RITC)-treated BV2 cells (Figures S23, S24). Moreover, previous study suggested that the migratory activity of microglia is regulated not only through gene expressions also can be regulated by activation/inactivation of intracellular signal molecules, including those regulating focal adhesion formation and actin re-organization [52,53]. The detailed activation/inactivation of intracellular signal molecules might be analysed with phosphoproteome analysis using LC-MS/MS in the further study.

Additional References

  1. Fan, Y.; Xie, L.; Chung, C.Y. Signaling Pathways Controlling Microglia Chemotaxis. Mol Cells. 2017, 40, 163-168.
  2. Franco-Bocanegra, D.K.; McAuley, C.; Nicoll, J.A.R.; Boche, D. Molecular Mechanisms of Microglial Motility: Changes in Ageing and Alzheimer's Disease. Cells. 2019, 8, 639.

  1. Regarding Figure 6D, the method for measuring intracellular ATP is presented only in supplementary information of the previous report, but not in this article. Please include it in this article. Also, regarding measurement of exocytosis, the method should be specified in section 2.11.

à As you point out, we added the method for measuring intracellular ATP and specified the title of section 2.11 (now 2.12).

Line 208, Page 5

2.11. Evaluation of intracellular ATP level

The ATP level of 0.01 and 0.1 µg/µl MNPs@SiO2(RITC)-treated BV2 cells were evaluated as previously described [35]. Briefly, relative levels of intracellular ATP were evaluated using the CellTiter-Glo® (Promega, Madison, WI, USA). The 0.01 and 0.1 μg/μl MNPs@SiO2(RITC)-treated BV2 cells were incubated at RT for 30 min, and luminescent reagent was added. The contents were mixed for 2 min and incubated for 10 min at RT. Luminescence signals were detected using a multi-mode microplate reader (SpectraMax® iD3; Molecular Devices, San Jose, CA, USA).

2.12. Flow cytometry for evaluation of relative intracellular NPs amount

BV2 cells were treated with 0.01 and 0.1 µg/µl of MNPs@SiO2(RITC) and silica NPs for 12 h and then treated citric acid for 12 h. The cells were trypsinised and washed twice with PBS and fixed in Cytofix buffer (BD, San Jose, CA, USA) at 4°C for 20 min. The cells were washed twice with PBS. The labelled cells were analysed using a flow cytometer (BD FACS Aria II™, BD, San Jose, CA, USA) at the 3D Immune System Imaging Core Facility of Ajou University.

  1. The authors conclude that recovery of BV2 migration by citrate is caused by stimulation of NP exocytosis. This is predicted from results of the omics analysis. However, the data in Figure 7B-E show only a small increase in NP release by citrate, and are not confirmed by statistical analysis. So, it is hard to conclude that citrate stimulates NP exocytosis based on these data. The possibility that citrate promotes NP exocytosis should be discussed only when statistically significant citrate-induced NP releases is obtained, after the same experiments are repeated.

à We agree with your concern, and it is technical limitation. Exocytosis analysis was performed with FACS several independent experiments and representative data was shown in 7B-E with arbitrary unit. FACS analysis is qualitative evaluation and huge inter-experimental variation of the fluorescence intensity is occurred every time, even in same instrument setting. Moreover, although we found NPs release in citrate and NPs treated cells, the exocytosis induction effect of citrate is very subtle (~10%). Thus, we could not deduce statistical significance for the experiment according to aforementioned factors and repetition cannot be a solution for the situation. We added these points to the revised manuscript as follows

Line 493, Page 14

An increment in intracellular ATP levels is significantly associated with the active transport of molecules, including exocytosis [43]. When citrate was added to MNPs@SiO2(RITC)-treated BV2 cells, the exocytosis of MNPs@SiO2(RITC) increased, which can be attributed to ATP acting as an energy source for NP exocytosis. The physicochemical properties of NPs, such as size, shape, and surface properties, can affect exocytosis [56,57]. Thus, further studies are needed to investigate the physicochemical properties of NPs and their exocytosis efficiency with or without citrate. In addition, citrate is a calcium chelator [58] and NPs increase cytosolic Ca2+ concentrations with Ca2+ pumps on the plasma membrane and the mitochondrial membrane, inducing Ca2+ toxicity in NP-treated cells [59]. Although we did not obtain direct evidence of calcium chelation induced by citrate, we hypothesized that citrate may act as a chelating agent of calcium produced by nanoparticles to reduce nanotoxicity. So far, due to the usefulness of NPs, compared to the studies examining endocytosis of NPs, fewer studies examining NP exocytosis have been published [43,60,61]. Therefore, further studies are needed to investigate the chelating effects of citrate in NP-treated cells. Moreover, we examined the NPs exocytosis by treatment of citrate using FACS in this study and the effect was too subtle for clear conclusion. Thus, detailed NPs exocytosis analysis is required using quantitative analytic method such as ICP-MS in the further study.

Reviewer 2 Report

In the present article Lee et al., and his coworkers studied the migratory activity of microglia after treating with NPs (with and without citrate). Authors utilized conventional biomolecular methods such as RNA-seq-based transcriptomics, and metabolomics analyses to evaluate the activation of microglia. Results revealed that migratory activity of microglia can be recovered by treatment with citrate. Overall, this study is useful to understand the importance of optimal concentrations of NPs and co-treated agents to further construct an efficient therapeutic/diagnostic agent for future brain disease applications. As a significant advance of the present article, I recommend it for publication after resolving the following minor issues.

1. Authors utilized concentrations of citrate is 1mM. What is the optimal concentration of citrate to recover the migratory activity of microglia? Is that 1 mM concentration is enough to recover? Is changing higher or lower the con of citrate may affect any?

2. Is that citrate capped NPs being good candidates for recover the migratory activity of microglia or need to utilize the external citrate agent?

3. The clear function of citrate is not clear in the present manuscript. Explain.

4. Cite the following papers in the introduction section, https://doi.org/10.3390/ijms23084153, https://doi.org/10.3390/polym12123055, https://doi.org/10.1016%2Fj.apsb.2020.11.023

5. Authors have utilized two concentration of NPs 0.1 µg/µl and 0.01 µg/µl for their study. From author point of view, is that using a 0.01 µg/µl dose of NPs good for future brain therapeutic applications?

Author Response

Answers to the comments of Reviewer #2

In the present article Lee et al., and his coworkers studied the migratory activity of microglia after treating with NPs (with and without citrate). Authors utilized conventional biomolecular methods such as RNA-seq-based transcriptomics, and metabolomics analyses to evaluate the activation of microglia. Results revealed that migratory activity of microglia can be recovered by treatment with citrate. Overall, this study is useful to understand the importance of optimal concentrations of NPs and co-treated agents to further construct an efficient therapeutic/diagnostic agent for future brain disease applications. As a significant advance of the present article, I recommend it for publication after resolving the following minor issues.

  1. Authors utilized concentrations of citrate is 1mM. What is the optimal concentration of citrate to recover the migratory activity of microglia? Is that 1 mM concentration is enough to recover? Is changing higher or lower the con of citrate may affect any?

à The citrate treatment concentration 1 mM for was determined as optimal and maximum concentration to alleviate MNPs@SiO2(RITC) induced toxicity in previous study (reference 42). In addition, we observed that cells cannot live in the citrate concentration over 1 mM according to low pH condition. We added these points to the revised manuscript as follows

Line 385, Page 11

3.6. Alleviation effect of citrate on the reduction of migratory activity in MNPs@SiO2(RITC)-treated BV2 cells

Among the triple omics networks, reduction in OA levels, which elucidate suppression of the tricarboxylic acid (TCA) cycle, was dominantly observed and highly related to other genes and proteins. We hypothesised that supplementation with OAs could be employed as a strategy for the alleviation of MNPs@SiO2(RITC)-induced migratory activity reduction. Among the OAs, citrate is a candidate for OA supplementation according to the expression of citrate transporters in the plasma membrane and mitochondrial membrane for proper delivery, and citrate treatment concentration 1 mM was determined as optimal and maximum concentration to alleviate MNPs@SiO2(RITC) induced toxicity in previous study [42]. Migratory activity in MNPs@SiO2(RITC)-treated BV2 cells was analysed in the presence of citrate (Figure 6A). The migratory activity was decreased in an MNPs@SiO2(RITC)-dose-dependent manner, and this decrease was alleviated in citrate co-treated BV2 cells (Figure 6B). No significant changes were observed in the cell viability in MNPs@SiO2(RITC)-treated BV2 cells or citrate co-treated BV2 cells (Figure 6C). However, intracellular ATP levels were reduced in MNPs@SiO2(RITC)-treated cells, and the intracellular ATP reduction was alleviated by citrate co-treatment (Figure 6D).

  1. Is that citrate capped NPs being good candidates for recover the migratory activity of microglia or need to utilize the external citrate agent?

à In our previous study, citrate was selected for alleviation of NPs induced toxicity by following expected effects: i) nutritional supplementation for TCA cycle intermediate; ii) chelating intracellular calcium ion and toxic NPs derived metal ion. As you mentioned, capped NPs or utilization the external citrate agent is great idea, and it might alleviate NPs induced toxicity. Moreover, the citrate has been used as NPs stabilizer by capping agent and storage formulation. In addition, to fulfill the effect i), the citrate should be release from the NPs and enter mitochondria. We added these points to the revised manuscript as follows

Line 448, Page 13

Citrate has been purposed in previous study as alleviation agent for nanomaterial induced toxicity by following effects: i) nutritional supplementation for TCA cycle intermediate; ii) chelating intracellular calcium ion and toxic NPs derived metal ion [35]. In this study, we hypothesized that ATP levels increase upon treatment with citrate in MNPs@SiO2(RITC)-treated BV2 cells with an increase in the migratory activity as citrate is converted into isocitrate via cis-aconitate by aconitase for the production of ATP in mitochondria [48]. In addition, citrate has been used as NPs stabilizer by capping agent and storage formulation [49-51]. Thus, conditionally releasable citrate capping or utilization of citrate as external agent might be strategy for alleviation of NPs induced toxicity.

Additional References

  1. Nigam, S.; Barick, K.C.; Bahadur, D. Development of citrate-stabilized Fe3O4 nanoparticles: Conjugation and release of doxorubicin for therapeutic applications. Journal of Magnetism and Magnetic Materials. 2011, 323, 237-243.
  2. Grys, D.B.; de Nijs, B.; Salmon, A.R.; Huang, J.; Wang, W.; Chen, W.H.; Scherman, O.A.; Baumberg, J.J. Citrate Coordination and Bridging of Gold Nanoparticles: The Role of Gold Adatoms in AuNP Aging. ACS Nano. 2020, 14, 8689-8696.
  3. Tran, M.; DePenning, R.; Turner, M.; Padalkar, S. Effect of citrate ratio and temperature on gold nanoparticle size and morphology. Materials Research Express. 2016, 3, 105027.

  1. The clear function of citrate is not clear in the present manuscript. Explain.

à As aforementioned, we added the description for expected effects of citrate as follows

Line 448, Page 13

Citrate has been purposed in previous study as alleviation agent for nanomaterial induced toxicity by following effects: i) nutritional supplementation for TCA cycle intermediate; ii) chelating intracellular calcium ion and toxic NPs derived metal ion [35]. In this study, we hypothesized that ATP levels increase upon treatment with citrate in MNPs@SiO2(RITC)-treated BV2 cells with an increase in the migratory activity as citrate is converted into isocitrate via cis-aconitate by aconitase for the production of ATP in mitochondria [48]. In addition, citrate has been used as NPs stabilizer by capping agent and storage formulation [49-51]. Thus, conditionally releasable citrate capping or utilization of citrate as external agent might be strategy for alleviation of NPs induced toxicity.

  1. Cite the following papers in the introduction section, https://doi.org/10.3390/ijms23084153, https://doi.org/10.3390/polym12123055, https://doi.org/10.1016%2Fj.apsb.2020.11.023

à As you recommended, we added the references with descriptions as follows

Line 49, Page 2

        NPs can penetrate the brain by passing the blood-brain barrier (BBB) through receptors and transporters expressed on the endothelial cells of brain capillaries, and the NPs have been used for one of the tools as brain-targeting diagnostics and drug delivery [11-13]. However, NPs internalised in the brain can induce cytotoxicity by oxidative stress, neuroinflammation, endoplasmic reticulum (ER) stress, disrupted signal pathways, and neurodegeneration in neurons [5,14]. Thus, the neurotoxicity of internalised NPs in the brain have been considered [15-19]. In addition, several issues have been noted in microglia exposed to nanoparticles, the primary immune cells in the brain, which migrate toward damaged regions and play a prominent role in neuroinflammation and neurodegeneration [20,21].

Additional References

  1. Han, L.; Jiang, C. Evolution of blood-brain barrier in brain diseases and related systemic nanoscale brain-targeting drug delivery strategies. Acta Pharm Sin B. 2021, 11, 2306-2325.
  2. Hersh, A.M.; Alomari, S.; Tyler, B.M. Crossing the Blood-Brain Barrier: Advances in Nanoparticle Technology for Drug Delivery in Neuro-Oncology. Int J Mol Sci. 2022, 23, 4153.
  3. Thangudu, S.; Cheng, F.Y.; Su, C.H. Advancements in the Blood-Brain Barrier Penetrating Nanoplatforms for Brain Related Disease Diagnostics and Therapeutic Applications. Polymers (Basel). 2020, 12, 3055.

  1. Authors have utilized two concentration of NPs 0.1 µg/µl and 0.01 µg/µl for their study. From author point of view, is that using a 0.01 µg/µl dose of NPs good for future brain therapeutic applications?

à Usage of NPs in brain might be inevitable due to their BBB penetration property. We previously calculated that theoretically 100 mg/kg MNPs@SiO2(RITC) dosage for mouse will localized a concentration of ~0.0125 µg/µl in the brain, and we observed microglial activation in the MNPs@SiO2(RITC) treated mouse brain at that dosage. Thus, our findings highlight the importance for minimization of dosages in NPs application of brain-targeting diagnostics and drug delivery to reduce possible adverse effects. We added these points to the revised manuscript as follows

Line 508, Page 14

In conclusion, the present study suggests that exposure to NPs might decrease the migratory activity of microglia, which can be recovered by citrate, based on a combination of conventional biomolecular methods, RNA-seq-based transcriptomics, and metabolomics analyses used to evaluate the activation of microglia treated with MNPs@SiO2(RITC). Although in vivo experiment will be direct way for showing the effect of MNPs@SiO2(RITC), the data obtained from BV2 cells, including integrated omics analysis, can be used to extrapolate the phenomena in in vivo. Our findings highlight the importance for minimization of dosages in NPs application of brain-targeting diagnostics and drug delivery to reduce possible adverse effects.  In addition, this study suggests that co-treatment with agents that reduce nanotoxic effects on microglia when NPs are used as therapeutic and diagnostic agents for brain diseases.

Round 2

Reviewer 1 Report

none